# Sodium Hexametaphosphate Serves as an Inducer of Calcium Signaling

**DOI:** 10.3390/biom13040577

**Published:** 2023-03-23

**Authors:** Daiki Katano, Woojin Kang, Yuichirou Harada, Natsuko Kawano, Mami Miyado, Takako Saito, Mio Fukuoka, Mitsutoshi Yamada, Kenji Miyado

**Affiliations:** 1Department of Life Sciences, School of Agriculture, Meiji University, 1-1-1 Higashimita, Kawasaki 214-8571, Japan; 2Department of Reproductive Biology, National Research Institute for Child Health and Development, 2-10-1 Okura, Setagaya 157-8535, Japan; 3Laboratory Animal Resource Center, Transborder Medical Research Center, Faculty of Medicine, University of Tsukuba, 1-1-1 Tennodai, Tsukuba 305-8575, Japan; 4Department of Molecular Pathology, Tokyo Medical University, 6-1-1 Shinjuku, Shinjuku 192-0397, Japan; 5Department of Food and Nutrition, Beppu University, 82 Kita-Ishigaki, Beppu 874-8501, Japan; 6Department of Molecular Endocrinology, National Research Institute for Child Health and Development, 2-10-1 Okura, Setagaya 157-8535, Japan; 7Department of Applied Life Sciences, Faculty of Agriculture, Shizuoka University, 836 Ohya, Shizuoka 422-8529, Japan; 8Shizuoka Institute for the Study of Marine Biology and Chemistry, Shizuoka University, 836 Ohya, Shizuoka 422-8529, Japan; 9Department of Obstetrics and Gynecology, Keio University School of Medicine, 35 Shinanomachi, Shinjuku 160-8582, Japan; 10Division of Diversity Research, National Research Institute for Child Health and Development, 2-10-1 Okura, Setagaya 157-8535, Japan

**Keywords:** hexametaphosphate, female reproduction, oocyte, parthenogenesis, calcium rise

## Abstract

In bacteria, polymers of inorganic phosphates, particularly linear polyphosphate, are used as alternative phosphate donors for adenosine triphosphate production. A six-chain form of sodium metaphosphate, sodium hexametaphosphate (SHMP), is believed to have no physiological functions in mammalian cells. In this study, we explored the possible effects of SHMP on mammalian cells, using mouse oocytes, which are useful for observing various spatiotemporal intracellular changes. Fertilization-competent oocytes were isolated from the oviducts of superovulated mice and cultured in an SHMP-containing medium. In the absence of co-incubation with sperm, SHMP-treated oocytes frequently formed pronuclei and developed into two-cell embryos owing to the increase in calcium concentration in the cytoplasm. We discovered an intriguing role for SHMP as an initiator of calcium rise in mouse oocytes, presumably in a wide variety of mammalian cells.

## 1. Introduction

Phosphate polymers, specifically polyphosphate (polyP), are ubiquitously present inside cells from bacteria to mammals. PolyP is a linear polymer composed of a thousand phosphates polymerized by high-energy phosphate bonds and has various biological functions, ranging from energy storage to stress response in bacteria [1,2,3,4] (Figure 1a).

In mammalian cells, polyP is widely distributed in the nucleus, plasma membrane, and various types of organelles, including the mitochondria, although the mammalian enzyme that synthesizes polyP is unclear [3]. Recent studies suggest that polyP contributes to various biological processes in mammalian cells. For example, platelet-specific organelles, that is, dense granules, are known to contain polyP [5]. Through its release from platelets into the blood during bleeding, polyP plays a role in the hemostatic pathway by activating hemostatic Factor V. Subsequently, polyP promotes thrombin synthesis and delays fibrinolytic reaction [5].

PolyP activates a signaling pathway important for mammalian cell proliferation, which is activated by a protein kinase, the mammalian target of rapamycin (mTOR) [6]. Further investigation has demonstrated that polyP, with an average chain length of 15 or longer, markedly enhances mTOR activation, which phosphorylates downstream molecules [7]. In transgenic mice expressing a polyP-degrading enzyme derived from budding yeast, the number of litters obtained by intercrossing male and female transgenic mice decreased [8]. This result implies that polyP may play a beneficial role in fertility.

Alternative reports have shown that metaphosphates, in linear or circular form, exist naturally [9], raising the possibility that metaphosphates, but not the debris of polyP, also possess biological activity. Sodium hexametaphosphate (SHMP), a metaphosphate, has applications in a wide variety of industries, including as an additive in foods and children’s drinks [10] (Figure 1a) [11]. SHMP is widely used as a deflocculant in the production of clay-based ceramic particles. It is also used as a dispersing agent to break down clay and other soil types for texture assessment [12]. SHMP is also used as an active anti-staining and tartar-preventing ingredient in toothpaste [13].

Accumulated evidence shows biological functions of phosphate polymers, especially polyP, in mammalian cells. On the other hand, a role of short phosphate polymers, including SHMP, is still unknown. Eggs are larger than other types of cells, making it easier to observe various spatiotemporal changes. Moreover, the use of eggs allows us to investigate the physiological significance of such changes. In the present study, we explored the effects of exogenous SHMP supplementation on mammalian cells, using mouse oocytes.

## 2. Materials and Methods

### 2.1. Reagents and Preparation of SHMP Solution

A set of polyPs of definite length (cat No. 638-51691) (FUJIFILM Wako Pure Chemical Corp., Tokyo, Japan) contained a short polyP with 14 polyphosphates (S), a middle polyP with 60 polyphosphates (M), and a long polyP with 130 polyphosphates (L). Extra-long polyP (ExtL) (cat No. BXP-C-700), composed of 700 polyphosphates, was purchased from Funakoshi Co., Ltd. (Tokyo, Japan). SHMP (cat No. 305553-25G) was purchased from Sigma-Aldrich Co., LLC. (St. Louis, MO, USA). SHMP was dissolved in H_2_O to 16.3 mM (10 mg/mL) and filtered with an 0.45 µm pore size filter and a 10 mL syringe. The prepared solution was stored at room temperature (RT) for 7 days until use and was used at a final concentration of 1.63 mM (1 mg/mL). The SHMP solution was inactive immediately after preparation. Therefore, the SHMP solution was incubated for 2 days and longer to see whether activity would begin to appear. We assume that the structure, shown in Figure 1a, will be formed and stabilized 7 days after preparation. Nuclei were counterstained with 4’, 6-diamidino-2-phenylindole (DAPI) (FUJIFILM Wako Pure Chemical Corp.).

### 2.2. Animals

All the mice were housed under specific pathogen-free conditions. Food and water were provided ad libitum. The animal experiments were performed in accordance with the principles and guidelines of the Care and Use of Laboratory Animals at the National Research Institute for Child Health and Development. The Animal Committee of the National Research Institute for Child Health and Development approved all experiments, including animal experiments (experimental number: 04-004).

### 2.3. Isolation of Germinal Vesicle (GV) Oocytes and Detection of Phospahate Polymers

To isolate GV oocytes, ovarian follicles were isolated from C57BL/6J female mice (8–12 weeks old, purchased from Japan SLC Inc., Shizuoka, Japan) 36 h after injecting pregnant mare serum gonadotropin (PMSG). GV oocytes and granulosa cells were retrieved by manual rupture of the ovarian follicles using a 21-gauge syringe. These cells were stained with DAPI at a final concentration of 50 μg/mL. Fluorescent images were captured at wavelengths ranging from 502 to 588 nm using a confocal microscope (LSM 510; Carl Zeiss, Jena, Germany).

### 2.4. Detection of Phosphate Polymers by Electrophoresis

Denaturing urea polyacrylamide gel electrophoresis (urea-PAGE) is a useful technique for separating single-stranded DNA, RNA fragments, and phosphate polymers. Phosphate polymers were stained and visualized using Toluidine Blue O (TBO) staining after electrophoresis. As reported previously [3], denaturing urea polyacrylamide gel (15%) was prepared as follows: 30% acrylamide/bisacrylamide solution (29:1, 6 mL) (Bio-Rad Laboratories, Inc., Hercules, CA, USA), urea (Sigma-Aldrich) (7M), 10× Tris-Borate-EDTA buffer (pH = 8.3, 1.2 mL) (Nacalai Tesque, Inc., Kyoto, Japan), N,N,N′,N′-Tetramethyl ethylenediamine (Nacalai Tesque) (10 μL), 10% ammonium persulphate solution (24 μL) (Nacalai Tesque), and up to 12 mL with H_2_O.

To prepare the samples, the phosphate polymers were precipitated using ethanol. First, 10 µL of each sample, 25 µL of 100% ethanol (FUJIFILM Wako Pure Chemical), and 1 µL of 3 M sodium acetate (Nippon Gene Co., Ltd., Toyama, Japan) were mixed by inversion. Centrifugation was then performed at 4 °C and 1500 rpm for 20 min. After centrifugation, the supernatant was discarded, 100 µL of 70% ethanol was added, and centrifugation was performed at 4 °C and 1500 rpm for 3 min. After centrifugation, the supernatant was again discarded, and the remaining ethanol was volatilized by air drying. Ten µL of formamide (Sigma-Aldrich) and 10× loading dye (Takara Bio Inc., Kyoto, Japan) was added to dried phosphate polymers. The prepared samples were heated at 85 °C for 2 min immediately before urea-PAGE.

After urea-PAGE, the gels were stained by immersing them in a TBO pre-staining solution containing 10% MeOH and 10% acetic acid (FUJIFILM Wako Pure Chemical) and shaking for 15 min. After the solution was discarded, the gels were immersed in a TBO solution containing 0.05% TBO, 25% MeOH, 5% glycerol (FUJIFILM Wako Pure Chemical), and 5% acetic acid and shaken for 10 min. Thereafter, the gels were immersed in a TBO-decolorizing solution containing 25% methanol, 5% glycerol, and 5% acetic acid.

### 2.5. Isolation of Metaphase II-Arrested (MII) Oocytes and Incubation in SHMP-Containing Medium

Cumulus–oocyte complexes (COCs) were isolated from the oviductal ampulla of superovulated C57BL/6J female mice (8–12 weeks old) 14–16 h after human chorionic gonadotropin (hCG) injection [14]. They were placed in a 30 µL drop of Toyoda-Yokoyama-Hoshi (TYH) medium covered with paraffin oil (Nacalai Tesque, Inc., Kyoto, Japan) equilibrated with 5% CO_2_ in air at 37 °C. To examine the effects of SHMP in the absence of sperm, oocytes were transferred to SHMP-containing TYH medium at a final concentration of 1.63 mM.

### 2.6. Observation of Nuclear and Mitochondrial Changes

To monitor nuclear and mitochondrial changes in oocytes during SHMP treatment, we used a transgenic mouse line (RBGS002 mice), as previously reported [15]. A promoter for human cytomegalovirus (CMV) early enhancer/chicken β-actin (CAG) promoter fused to a mitochondrial import sequence (Atp5g1) drives the expression of RFP (DsRed2) in a wide variety of cells, including oocytes.

COCs were isolated from the oviductal ampulla of superovulated RBGS002 female mice (8–12 weeks old) 14–16 h after hCG injection. Oocytes were stained to monitor nuclear changes with DAPI at a final concentration of 1 μg/mL. To examine the effects of SHMP, oocytes were transferred to TYH medium containing SHMP at a final concentration of 1.63 mM.

### 2.7. Transmission Electron Microscopy (TEM)

Oocytes were isolated from superovulated female mice and incubated in SHMP-containing TYH medium at a final concentration of 1.63 mM. After 4 h of incubation, SHMP-treated oocytes were fixed with 2% glutaraldehyde in 0.1 M phosphate buffer (pH 7.4) overnight at RT and then washed thrice with 0.1 M phosphate buffer for 30 min. The oocytes were subjected to post-fixation with 2% osmium tetroxide in 0.1 M phosphate buffer at 4 °C for 1 h. The samples were dehydrated in graded ethanol solutions (50%, 70%, 89%, and 100%), transferred to a resin (Quetol-812; Nisshin EM Co., Tokyo, Japan), and polymerized for 48 h at 60 °C. The polymerized resins were ultrathin-sectioned at 70 nm with a diamond knife using an ultramicrotome (Ultracut UCT; Leica, Vienna, Austria) and mounted on copper grids. Sections were stained with 2% uranyl acetate for 15 min at RT, washed with distilled water, and subjected to secondary staining with a lead stain solution (Sigma-Aldrich) for 3 min at RT. The grids were observed under a transmission electron microscope (JEM-1400Plus; JEOL Ltd., Tokyo, Japan). We counted the number of mitochondria in the oocyte cytoplasm.

### 2.8. Measurement of Intracellular Ca^2+^ Concentration

To monitor the intracellular Ca^2+^ concentration, oocytes were incubated in TYH medium containing the Ca^2+^-sensitive fluorescent dye Oregon green 488 BAPTA-1 AM (final concentration of 2 μM, Molecular Probes, Invitrogen, Carlsbad, CA, USA) for 15 min at 37 °C in a CO_2_ incubator. The oocytes were then washed three times with TYH medium (5 min per wash). Oocytes were incubated in SHMP-containing TYH medium at a final concentration of 1.63 mM.

Fluorescence images were captured every 10 s using a highly sensitive CCD camera (Andor Technology, Belfast, UK) with software to operate the camera (Yokogawa, Tokyo, Japan). The fluorescence intensity was measured using the Andor IQ imaging software (version 1.10.1) (Andor Technology). The fluorescence intensity of individual oocytes was measured within a user-selected region covering most of the oocyte area. The mean intensity over the same area for each image in the time series was automatically analyzed.

To examine the effect of oligoDNA on Ca^2+^ concentration, the oocytes were incubated in TYH medium containing 6-mer oligoDNA mixture at a final concentration of 1.8 mM (0.45 mM each of hexaATP, hexaCTP, hexaGTP, and hexaTTP).

### 2.9. Statistical Analysis

Significant differences (*p*-values) were calculated using Student’s *t*-test, and statistical significance was defined as *p* < 0.05. Results are expressed as mean ± standard error (SE).

## 3. Results

### 3.1. Presence of Phosphate Polymers in the Outer Region of Mouse Oocytes

As reported previously [16], phosphate polymers bind with DAPI at a high concentration, and its cellular distribution can be seen visually. The presence of phosphate polymers has been reported in mature mouse oocytes (metaphase II-arrested oocytes; MII oocytes), more specifically, in the extracellular coat surrounding the oocyte, the zona pellucida [8]. Similarly, when premature mouse GV oocytes were stained with DAPI, phosphate polymers were detected (Figure 1b,c and Appendix A). When observed more carefully, the minor population largely stored phosphate polymers inside the cells (arrows in Figure 1c). However, the fate of oocytes that store phosphate polymers in the cytoplasm remains unknown.

### 3.2. Electrophoretic Analysis of Phosphate Polymers

Because electrophoretic analysis requires extracts from a large number of cells, the size of polyP in mouse oocytes is yet to be determined. In contrast, phosphate polymers, categorized as polyP with various sizes, are commercially available, and electrophoretic analysis can be applied to them. To measure their lengths, electrophoretic analysis was performed (Figure 1d). When four materials, short-sized polyP with 14 polyphosphates (S), middle-sized polyP with 60 polyphosphates (M), long-sized polyP with 130 polyphosphates (L), and extra-long-sized polyP (ExL), were electrophoresed in a denaturing urea polyacrylamide gel, they were detected as broad bands. Similarly, SHMPs were detected as broad bands.

### 3.3. Influence of SHMP on Mouse Oocytes before Fertilization

To examine the possible function of SHMP, we cultured mouse oocytes in SHMP-containing fertilization medium (hereafter referred to as medium) (Figure 2a). MII oocytes were isolated from the oviducts of superovulated C57BL/6J female mice. When mouse oocytes were incubated in SHMP-containing medium for 4 h, the oocyte cytoplasm was enlarged, and there were several oocytes without the cavity, which is the space between the oocyte cytoplasm and the zona pellucida. SHMP treatment appeared to increase the oocyte cytoplasm (Figure 2b) and induce cytoplasmic vibration (Appendix A). In fact, the diameter was significantly more enhanced in SHMP-treated oocytes (88.6 ± 1.3 μm) than in untreated oocytes (72.6 ± 0.5 μm; *p* < 0.0001). Based on these results, we hypothesized that SHMP affects the oocytes.

### 3.4. Partial Parthenogenesis in SHMP-Treated Oocytes

As shown in Figure 3a, we examined the effects on the developmental ability of MII oocytes. When oocytes were observed at 6 and 24 h, pronuclei and two-cell embryos were often observed (Figure 3b and Appendix A). Pronuclear formation occurred in 4 of the 79 oocytes treated with SHMP (5.1%) (Figure 3c) compared to 0 of the 77 untreated oocytes (0.0%). Further, 2-cell embryos were also observed in 8 out of 79 oocytes (10.1%) (Figure 3d), compared to 0 out of 77 untreated oocytes (0.0%). However, the SHMP-treated oocytes did not develop into four-cell embryos. This result reinforces the hypothesis that SHMP moves the oocyte machinery related to developmental ability in the absence of sperm fusion.

### 3.5. Mitochondrial Alteration in SHMP-Treated Oocytes

To explore structural changes in SHMP-treated enlarged oocytes, MII oocytes were analyzed histologically. Since mitochondria are thought to be ancestral to aerobic bacteria, we raised the possibility that mitochondria could independently utilize exogenous SHMP to resume metaphase-II arrested in oocytes.

To monitor mitochondrial changes in oocytes during SHMP treatment, we used a transgenic line of mice (RBGS002 mice) [15]. As depicted in Figure 4a, MII oocytes were isolated from superovulated RBGS002 and C57BL/6J mice and incubated for 4 h in the SHMP-containing medium. Nuclei were stained with DAPI. Subsequently, mitochondrial localization was noticeably unaltered between SHMP-treated and untreated oocytes from RBGS002 mice. Instead, the nucleus was often localized closer to the center of the oocytes (Figure 4b,c). The distance from the center of the oocytes to the nucleus was significantly narrower in SHMP-treated oocytes than in untreated oocytes (*p* < 0.0012).

### 3.6. Ultrastructural Analysis of Mitochondria

We examined the mitochondria of MII oocytes isolated from C57BL/6J mice using electron microscopy. When the oocytes were observed, round internal structures with a high electron density (hereafter, round bodies) were detected both in SHMP-treated oocytes and untreated oocytes (Figure 4d). The percentage of oocytes carrying mitochondria with round bodies was comparable between SHMP-treated and untreated oocytes (Figure 4e). Otherwise, the number of round bodies per mitochondrion was significantly more increased in SHMP-treated oocytes (1.01 ± 0.07) than in untreated oocytes (0.59 ± 0.03; *p* < 0.0024) (Figure 4f). Based on this result, we assumed that the physiological features would be altered in the mitochondria of SHMP-treated oocytes. Furthermore, the length of microfilaments was short in SHMP-treated oocytes (368.0 ± 32.1 nm), compared with untreated oocytes (545.2 ± 45.7 nm; *p* < 0.0023) (Figure 4g,h).

### 3.7. Calcium (Ca^2+^) Concentration in SHMP-Treated Oocytes

In oocytes, Ca^2+^ is repeatedly elevated upon fertilization with sperm (Ca^2+^ oscillation), triggering intracellular changes necessary for subsequent development [17]. In neurons, polyP binds to receptors and induces Ca^2+^ release from the endoplasmic reticulum, leading to the activation of neural activity [18]. As shown in Figure 3, SHMP treatment of MII oocytes, albeit in small numbers, resulted in the production of two-cell embryos. Therefore, we examined the possibility that Ca^2+^ oscillation or Ca^2+^ increase also occurred in SHMP-treated oocytes. As depicted in Figure 5a, MII oocytes were isolated from the oviducts of superovulated C57BL/6J female mice and incubated in the SHMP-containing medium. To monitor the intracellular Ca^2+^ concentration, oocytes were incubated in TYH medium containing the Ca^2+^-sensitive fluorescent dye Oregon green 488 BAPTA-1 AM.

When images were serially captured, the fluorescence intensity of the Ca^2+^ indicator gradually increased in SHMP-treated oocytes, but not in untreated oocytes (Figure 5b). The relative fluorescence intensity (1.0 set at 0.0 min) more gradually increased in SHMP-treated oocytes than in untreated oocytes (Figure 5c). At 120 min during SHMP treatment, the intensity was significantly more enhanced in SHMP-treated oocytes (2.23 ± 0.31) than in untreated oocytes (1.49 ± 0.05; *p* < 0.0083).

Since nucleotides are phosphate polymers, oligonucleotides likely induce a Ca^2+^ increase in MII oocytes. To test this possibility, the oocytes were incubated in the medium containing a 6-mer oligoDNA mixture at a final concentration of 1.8 mM (0.45 mM each of hexaATP, hexaCTP, hexaGTP, and hexaTTP) (Figure 6a). Unfortunately, no Ca^2+^ increase was observed in oocytes treated with the 6-mer oligoDNA mixture (Figure 6).

## 4. Discussion

In this study, we explored the effects of exogenous SHMP supplementation on mouse oocytes (Figure 7). Our results demonstrated that (1) SHMP significantly increased the mass of the oocyte cytoplasm; (2) SHMP caused partial parthenogenesis, leading to pronuclear and two-cell formation; (3) SHMP altered the localization of nuclei in oocytes and affected the internal structure of mitochondria; (4) SHMP serially increased the Ca^2+^ concentration, but not Ca^2+^ oscillation. Collectively, these results suggest that SHMP enhances Ca^2+^ signaling in mammalian cells, providing mechanistic insights into the biological functions of phosphate polymers as ancient energy metabolites.

### 4.1. Transient Ca^2+^ Rise and Developmental Ability

In mammals, sperm fusion and Ca^2+^ oscillation promote complete development of fertilized oocytes [19,20]. Specific sperm-derived factors are believed to induce Ca^2+^ oscillation. This concept led to the discovery of a sperm factor, phospholipase C zeta, which functions widely in Ca^2+^ oscillation in mammals and birds [21]. However, sperm-abundant, but not sperm-specific, factors also contribute to oocyte activation after sperm fusion in vertebrates [20,22]. A transient Ca^2+^ rise is employed in the mechanism underlying oocyte activation in some non-mammalian vertebrates [23,24] and insects [25]. The present study found that SHMP caused an increase in intracellular Ca^2+^. In terms of insect repellency, the induction of parthenogenesis leads to the birth of an infertile subsequent generation.

### 4.2. Physiological Functions of PolyP

Apart from sperm–egg fusion, the increase in the intracellular Ca^2+^ concentration plays an important role as a signaling factor. In vitro experiments using neurons have shown that polyP functions as a signal transducer in the glia by binding to the P2Y1 receptor, a purine receptor whose ligands include ADP, releasing intracellular Ca^2+^ and activating phospholipase C [26]. Furthermore, it has been shown that the addition of polyP to primarily cultured neuroglial cells induces the release of endogenous polyP from astrocytes and promotes the uptake of extracellular polyP in neurons [27]. In the present study, we first discovered the role of SHMP in the intracellular Ca^2+^ rise (Figure 5).

As previously reported [28], polyP binds to unfolded proteins with high affinity in an ATP-independent manner in vitro. Moreover, polyP stabilizes proteins in vivo and protects them against stress-induced unfolding and aggregation, suggesting that polyP serves as a chaperone.

### 4.3. Physiological SHMP Activity

As described previously [29], the effects of two phosphate polymers, sodium triphosphate (STP) and SHMP, on the osteoblastic differentiation of human periodontal ligament cells and osteoblasts and bone formation were investigated. Both STP and SHMP increased the phosphorylation of adenosine monophosphate-activated protein kinase, Akt, mTOR, and mitogen-activated protein kinases. STP, but not SHMP, significantly increased new bone formation in mice. Collectively, these results suggested that STP and SHMP promoted osteoblastic differentiation in vitro, whereas STP stimulated bone repair in mice.

Morphological changes in the mitochondria were observed along with morphological changes in mouse oocytes (Figure 2 and Figure 4). It has been shown that the mitochondrial structure and amount of mitochondrial DNA (mtDNA) in oocytes change with age. Simsek-Duran et al. [30] reported that the amount of mtDNA in old mouse oocytes was significantly lower than that in young mice. The mitochondrial structure and number of mitochondria with internal cavities, such as those increased by SHMP treatment in the present study (Figure 4d), will increase. Therefore, it is possible that, in the presence of SHMP, there may be changes in fertilization and development due to mitochondrial abnormalities.

### 4.4. Possible Mechanism of SHMP-Triggered Events

PolyP can evoke Ca^2+^ release from internal stores through P2Y1, a G protein-coupled purinergic receptor [31]. P2Y1 belongs to the P2Y family, whose endogenous ligands are extracellular purine nucleotides (ATP and adenosine diphosphate) and pyrimidine nucleotides (uridine triphosphate [UTP] and uridine diphosphate). The P2Y family upregulates responses to stress or injury and mediates tissue regeneration in a variety of tissues through activating multiple signaling pathways [32]. P2Y is classified into eight subtypes (P2Y1, P2Y2, P2Y4, P2Y6, P2Y11, P2Y12, P2Y13, and P2Y14). P2Y2 mRNAs are expressed in bovine oocytes and cumulus cells [33]. Stimulation with UTP leads to a Ca^2+^ increase in both types of cells, and this effect is blocked by a P2Y2 inhibitor, suramin [34]. P2Y2 are also expressed in mouse oocytes and cumulus cells [33]. From these reports, we suppose that the P2Y family serves as receptors for phosphate polymers.

In fertilization, sperm fusion evokes Ca^2+^ oscillation, and the size of oocytes is unaltered. In contrast, since SHMP treatment enhanced the cell size of mouse oocytes (Figure 2), SHMP is considered to induce distinct signaling(s) from fertilization. In general, the high-water permeability of cell membranes causes rapid osmotic responses via the movement of osmotically active ions into or out of cells. Hypertrophy of the cell volume is observed when signaling from P2Y2 is transduced in muscles [35].

Since microtubules and microfilaments are connected with intracellular organelles, these alterations lead to the altered distribution of intercellular organelles. Ca^2+^-dependent microfilament reformation controls the movement of intracellular organelles [36]. Ultimately, cytoskeletons work as an actor in the “upstream or downstream” of Ca^2+^ signaling. Several research studies suggest that cell volume alterations result in cytoskeletal rearrangement [37].

The partial parthenogenic development of SHMP-treated eggs may be explained by changes in microfilaments. Since SHMP-treated eggs showed no release of a second polar body, the formation of two pronuclei is considered to originate from the unreleased polar body (Figure 3b). Since microfilaments are located near MII chromosomes and are involved in the polar body release [38], its deformation may disturb the movement of MII chromosomes.

A shown in Figure 5 and Figure 6, SHMP, but not oligoDNA, induced various changes in mouse eggs. There are no results yet that can explain the difference between the two. However, it is possible that SHMP forms some structure, since it takes at least 48 h for SHMP to show activity.

### 4.5. Phosphate World Inside and Outside Organisms

As recently reported [39], biochemical and biophysical examinations have revealed that soluble forms of phosphate species are more complex than previously known. Hence, phosphate polymers, including polyP and SHMP, can be present both physiologically and synthetically inside and outside cells and probably in seawater.

No previous studies have shown that exogenous SHMP affects mitochondria in mammalian oocytes. In the future, it will be necessary to examine the effects of exogenous SHMP not only in mouse oocytes, but also in various types of mammalian cells, to analyze the overall effects on Ca^2+^ signaling.

Our study was conducted in a culture system and did not show the effects of in vivo administration. Therefore, even if SHMP is contained in various foods and drinks, we cannot say that similar effects are induced in the body.

## Figures and Tables

**Figure 1 biomolecules-13-00577-f001:**
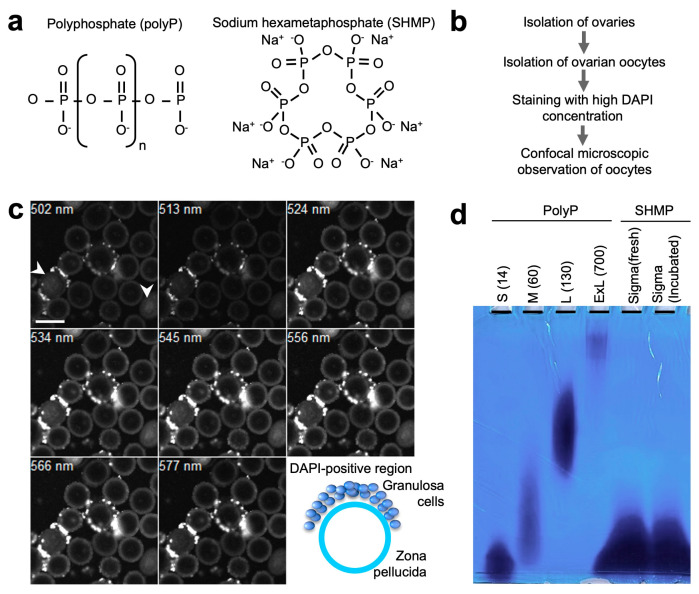
Detection of phosphate polymers by DAPI staining and electrophoresis. (**a**) Molecular formulae of phosphate polymers with a liner form (polyP) and with a circular form (SHMP). (**b**) Experimental flow. GV-staged oocytes with granulosa cells were isolated from ovaries of C57BL/6J female mice 36 h after PMSG treatment. After DAPI staining, fluorescence images were captured with wavelength from 502 to 577 nm. (**c**) Distribution of phosphate polymers in GV-staged oocytes and granulosa cells. A schematic drawing exhibits DAPI-stained (blue-colored) regions. Scale bar, 100 μm. (**d**) Electrophoresis of phosphate polymers, polyP, and SHMP. After electrophoresis, phosphate polymers were stained with TBO. S (14) (polyP with 14-mer short length), M (60) (polyP with 60-mer medium length), L (130) (polyP with 130-mer long length), and ExL (700) (polyP with 700-mer extraordinary long length) were applied. SHMP (fresh) and SHMP (incubated in RT for 7 days) were also applied.

**Figure 2 biomolecules-13-00577-f002:**
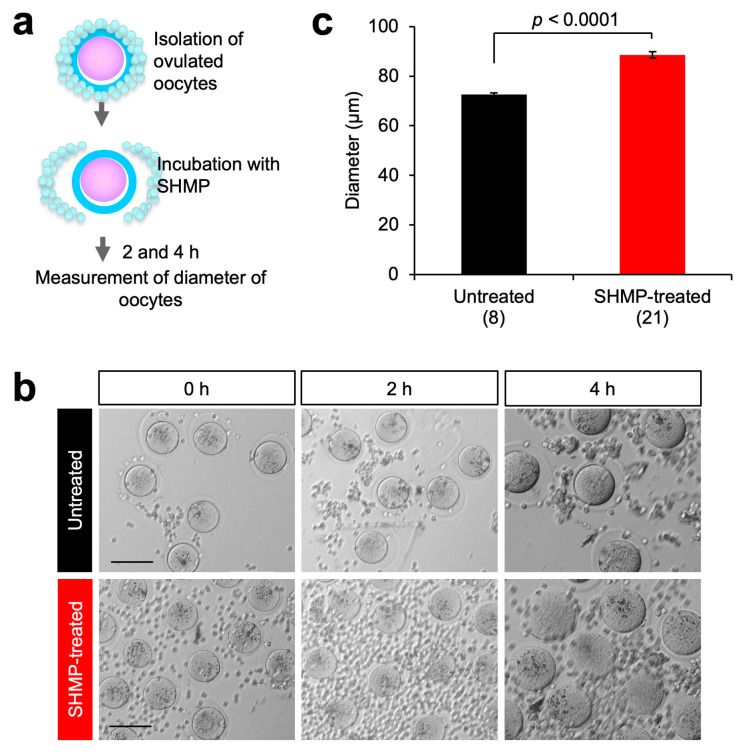
Treatment of ovulated oocytes with SHMP. (**a**) Experimental flow. COCs were isolated from the oviduct of superovulated C57BL/6J female mice and then treated with hyaluronidase. After washing, oocytes were incubated in the medium containing SHMP. The oocytes were observed 2 and 4 h after SHMP treatment. (**b**) Bright-field images of the oocytes after SHMP treatment. Scale bars, 100 μm. (**c**) Diameter of the oocytes treated with SHMP 4 h after SHMP treatment. Values are expressed as means ± SE. Parentheses indicate the number of oocytes examined.

**Figure 3 biomolecules-13-00577-f003:**
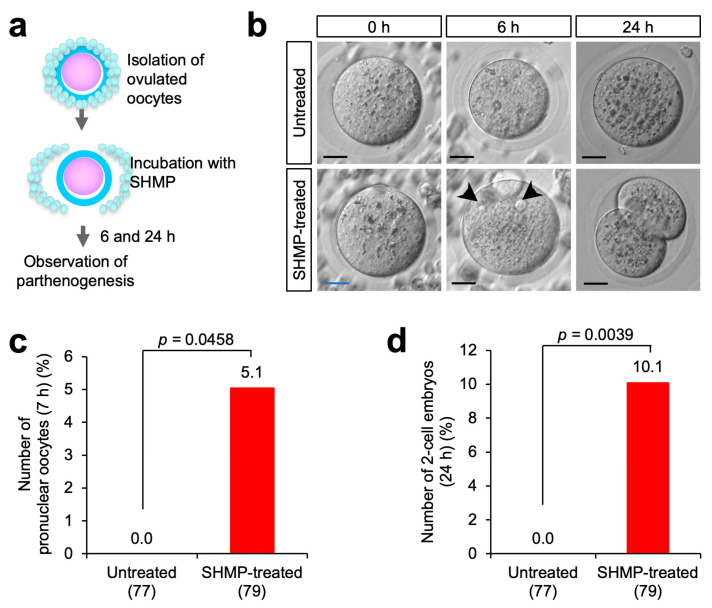
Partial parthenogenesis in oocytes treated with SHMP. (**a**) Experimental flow. COCs were isolated from the oviduct of superovulated C57BL/6J female mice and then treated with hyaluronidase. After washing, oocytes were incubated in the medium containing SHMP. The oocytes were observed 6 and 24 h after SHMP treatment. (**b**) Bright-field images of oocytes after SHMP treatment. Arrowheads indicate pronuclei. Scale bars, 20 μm. (**c**) Number of pronuclear oocytes (%). (**d**) Number of 2-cell embryos (%). Parentheses indicate the number of oocytes examined.

**Figure 4 biomolecules-13-00577-f004:**
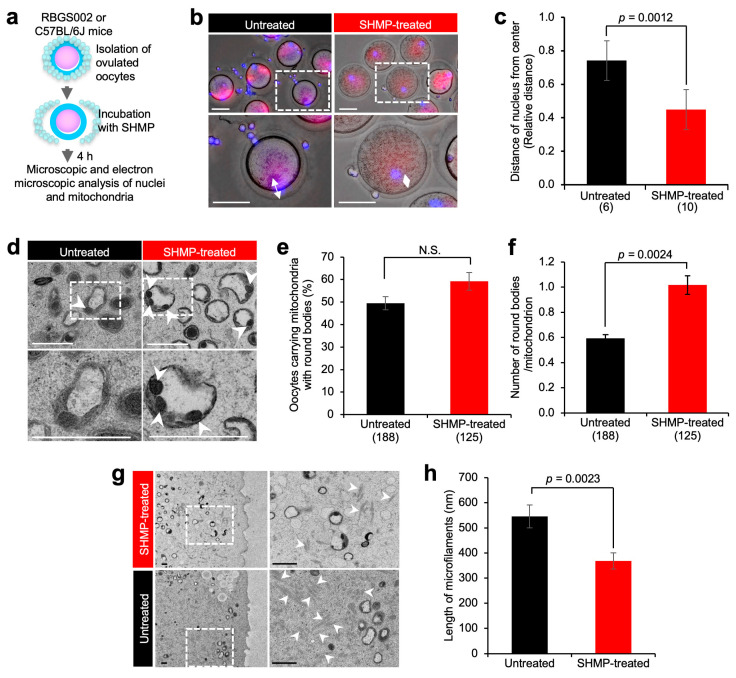
Nuclear and mitochondrial localization in ovulated oocytes after SHMP treatment. (**a**) Experimental flow. COCs were isolated from the oviduct of superovulated RBGS002 female mice that express DsRed2 in mitochondria. After hyaluronidase treatment and washing, the distance from nuclei to the center of oocytes was measured 4 h after SHMP treatment. Oocyte nuclei were counterstained with DAPI. COCs were also isolated from superovulated C57BL/6J female mice. After hyaluronidase treatment and washing, the oocytes were subjected to TEM analysis. (**b**) Nuclei of the oocytes. Dotted boxes are enlarged in lower images. Double arrows indicate the distance from the nucleus to the center of an oocyte. Scale bar, 50 μm. (**c**) The distance from the nuclei to the center of the oocytes. (**d**) TEM images of mitochondria. Dotted boxes are enlarged in lower images. Arrowheads indicate round bodies. Scale bar, 200 nm. (**e**) Percentage of mitochondria with round bodies (%). (**f**) Number of round bodies per mitochondrion. N.S., no significance. (**g**) TEM images of microfilaments. Dotted boxes are enlarged in right images. Arrowheads indicate microfilaments. Scale bars, 0.5 μm*. (***h**) The length of microfilaments (nm). Values are expressed as means ± SE. Parentheses indicate the number of oocytes examined.

**Figure 5 biomolecules-13-00577-f005:**
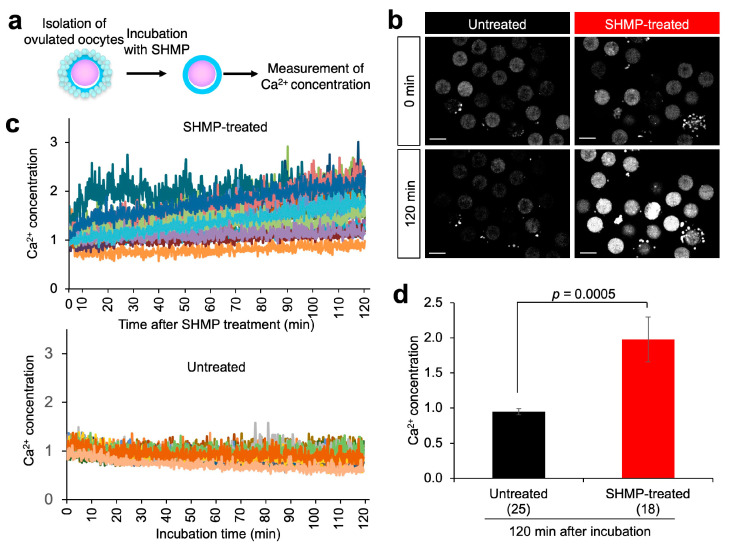
Ca^2+^ concentration in oocytes exposed to SHMP. (**a**) Experimental flow. COCs were isolated from the oviduct of superovulated C57BL/6J female mice and then treated with hyaluronidase. After washing, oocytes were incubated in the medium containing Oregon Green 488 BAPTA-1 AM for 15 min. The oocytes were next incubated in the medium containing SHMP, and their Ca^2+^ concentration was serially recorded. (**b**) Fluorescence images of oocytes 0 and 120 min after SHMP incubation. Scale bars, 80 μm. (**c**) Fluorescence intensity of Ca^2+^ concentration from 0 min to 120 min after SHMP treatment. SHMP-treated oocytes (n = 18) and untreated oocytes (n = 25) were examined. (**d**) Fluorescence intensity of Ca^2+^ concentration 120 min after SHMP treatment. Values are expressed as means ± SE. Parentheses indicate the number of oocytes examined.

**Figure 6 biomolecules-13-00577-f006:**
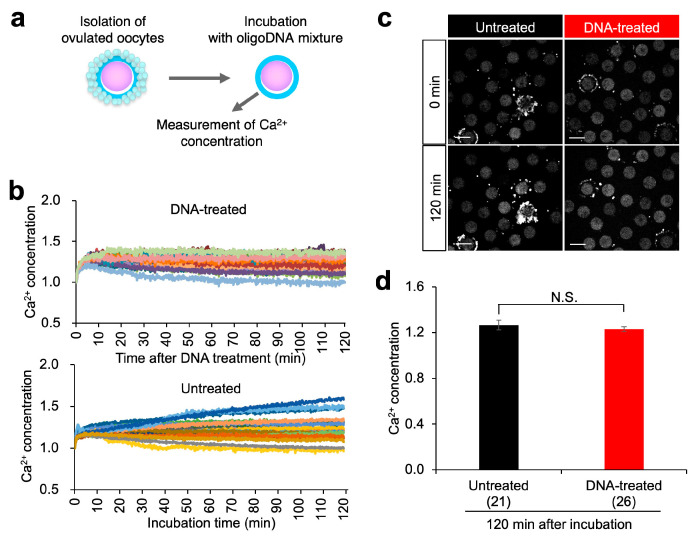
Ca^2+^ concentration in oocytes exposed to 6-mer oligoDNA mixture. (**a**) Experimental flow. COCs were isolated from the oviduct of superovulated C57BL/6J female mice and then treated with hyaluronidase. After washing, oocytes were incubated in the medium containing Oregon Green 488 BAPTA-1 AM for 15 min. The oocytes were next incubated in the medium containing oligoDNA mixture, and their Ca^2+^ concentration was serially recorded. (**b**) Fluorescence images of oocytes 0 and 120 min after oligoDNA incubation. Scale bars, 80 μm. (**c**) Fluorescence intensity of Ca^2+^ concentration from 0 min to 120 min after oligoDNA treatment. The oligoDNA-treated oocytes (n = 26) and untreated oocytes (n = 21) were examined. (**d**) Fluorescence intensity of Ca^2+^ concentration 120 min after oligoDNA treatment. Values are expressed as means ± SE. Parentheses indicate the number of oocytes examined. N.S., no significance.

**Figure 7 biomolecules-13-00577-f007:**
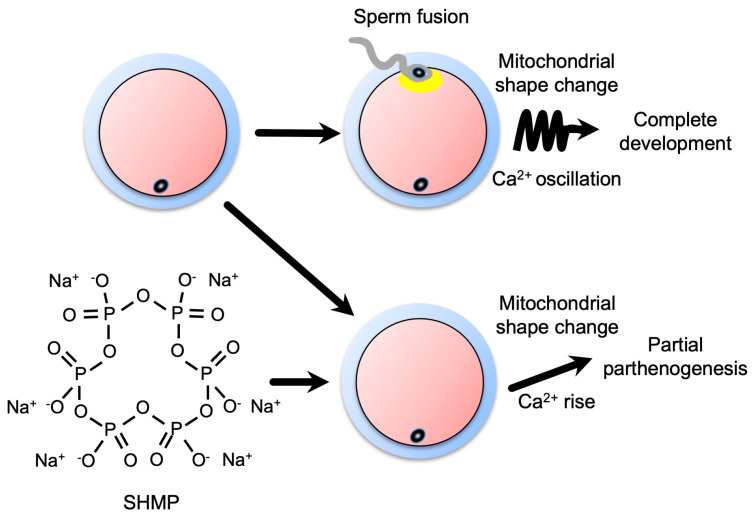
Schematic model of the effect of SHMP on oocytes. After sperm fusion, Ca^2+^ oscillation is induced in oocytes. Simultaneously, the mitochondrial inner structure is altered. In contrast, Ca^2+^ rise, but not oscillation, occurs in oocytes exposed to SHMP. Concurrently, the mitochondrial inner structure is altered.

## Data Availability

The data supporting the findings of this study are available from the corresponding author upon request.

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
