# Peer review of "Sodium Hexametaphosphate Serves as an Inducer of Calcium Signaling"

_biomolecules, 2023, doi:10.3390/biom13040577_

Round 1

Reviewer 1 Report

Manuscript biomolecules-2265540 – “Phosphate polymer consumed as a food additive induces calcium rise in mouse oocytes”

GENERAL COMMENTS

The manuscript is focused on the effects of the polymer SHMP, a food additive, on murine oocytes. SHMP was supplemented to in vitro culture media, and effects on oocytes (e.g., presence of polymer, oocyte size, development, mitochondria and Ca2+ concentration) were investigated. Differences were found compared to untreated oocytes.

The topic is novel, interesting, and worth investigation. The study design (in vitro addition of SHMP), however, does not resemble the actual, in vivo exposure to SHMP, and so the conclusions cannot be extended to female reproduction in general. This should be clarified in the manuscript, and the sentence “In this study, we explored the possible effects of SHMP on mammalian cells, focusing on female reproduction” should be rephrased.

In my opinion, the manuscript is acceptable for publication after minor revisions.

SPECIFIC COMMENTS

Title

The title might be misleading, in my opinion. The compound that is going to be tested is not specified. Furthermore, reading the title alone, I thought the present study tested its addition to food/feed, while it is an in vitro experiment. Thus, the title needs rephrasing.

M&M

I find the text a bit confusing. In the materials and methods section, the experimental design is not clear, but then methods and design are basically re-explained in the result section. Please move information to the relevant paragraphs.

Results

I can see nothing in Supplemental Figure S1, I can’t understand where the polymers are.

Line 230: Please remove beneficial or explain why they are considered so.

Figure 2b: Please explain why residual cumulus cells were left in culture with the oocytes. Could their presence influence the results of incubation with SHMP?

Paragraph 3.4: Statistical analysis for parthenogenesis rates is missing. Without statistical analysis it cannot be stated in the discussion/conclusion (Lines 334-335) that “SHMP caused partial parthenogenesis, leading to pro-nuclear and two-cell formation;”.

Figure 4a: There is a typo (“OF nuclei”).

Author Response

Reviewer #1:

Comment 1: The manuscript is focused on the effects of the polymer SHMP, a food additive, on murine oocytes. SHMP was supplemented to in vitro culture media, and effects on oocytes (e.g., presence of polymer, oocyte size, development, mitochondria and Ca2+ concentration) were investigated. Differences were found compared to untreated oocytes. The topic is novel, interesting, and worth investigation.

Response: First of all, we appreciate your thoughtful comment.

Comment 2: The study design (in vitro addition of SHMP), however, does not resemble the actual, in vivo exposure to SHMP, and so the conclusions cannot be extended to female reproduction in general. This should be clarified in the manuscript, and the sentence “In this study, we explored the possible effects of SHMP on mammalian cells, focusing on female reproduction” should be rephrased. In my opinion, the manuscript is acceptable for publication after minor revisions.

Response: Thank you for your valuable comment. As the reviewer pointed out, we have changed the previous sentence to the followings:

Eggs are larger than other types of cells, making it easier to observe various spatiotemporal changes. Moreover, the use of eggs allows us to investigate the physiological significance of such changes. In the present study, we explored the effects of exogenous SHMP supplementation on mammalian cells, using mouse oocytes (lines 66 – 69).

Comment 3: <Title> The title might be misleading, in my opinion. The compound that is going to be tested is not specified. Furthermore, reading the title alone, I thought the present study tested its addition to food/feed, while it is an in vitro experiment. Thus, the title needs rephrasing.

Response: As the reviewer pointed out, we have changed the previous title to “Sodium hexametaphosphate serves as an inducer of calcium signaling”.

Comment 4: <M&M> I find the text a bit confusing. In the materials and methods section, the experimental design is not clear, but then methods and design are basically re-explained in the result section. Please move information to the relevant paragraphs.

Response: Thank you for your valuable comment. We improved the results, and materials and methods sections.

Comment 5: <Results> I can see nothing in Supplemental Figure S1, I can’t understand where the polymers are.

Response: Thank you for your useful comment. To be able to check the eggs, we have added images with high contrast and brightness as Figure S1b.

Comment 6: Line 230: Please remove beneficial or explain why they are considered so.

Response: Accordingly, we have removed the word “beneficial”.

Comment 7: Figure 2b: Please explain why residual cumulus cells were left in culture with the oocytes. Could their presence influence the results of incubation with SHMP?

Response: We raised the possibility that SHMP acts on oocytes through cumulus cells or cumulus cell-derived substances. Therefore, to monitor both oocytes and cumulus cells, they were co-incubated, but the calcium concentration in the cumulus cells was unaltered.

Comment 8: Paragraph 3.4: Statistical analysis for parthenogenesis rates is missing. Without statistical analysis it cannot be stated in the discussion/conclusion (Lines 334-335) that “SHMP caused partial parthenogenesis, leading to pro-nuclear and two-cell formation;”.

Response: We apologize for not having performed the statistical analysis. Accordingly, we have performed the statistical analysis and added p-values in Figure 3c, d.

Comment 9: Figure 4a: There is a typo (“OF nuclei”).

Response: Accordingly, we have corrected the typo.

Reviewer 2 Report

In this manuscript, the author explored the functional effect of exogenous SHMP on female reproduction, especially on oocyte maturation and development. The results showed that SHMP could serve as an initiator of calcium rise and effect the fertilization process in mouse oocytes in vitro. The following are some concerns that need to be clarified.

1. For line 67, it is obscured and confusing for the readers to understand. What detailed effect did the author explore about exogenous SHMP supplementation? More clarifications should be addressed at the beginning (introduction part) to clearly state the scientific question of the whole manuscript.

2.Line 77, may I know the reason that prepared solution need to be stored at RT for 7 days until use? Is there any purpose to deliberately do that?

3. Since the SHMP caused partial parthenogenesis, did the author tested the location of the spindles? Was that also be affected?

4. The paper only listed several phenotypes after the exogenous SHMP supplementation, no detailed mechanism was explored based on these findings. The author should address the limitation deeply in the discussion part for this whole manuscript.

5. I recommend the author to reconsider and adjust the title. Since there is no necessary to mention the “food additive” based on this scientific question. What’s more, more instructible information should be shown directly in the title, to help readers understand the scientific issue directly and clearly.

6. The whole manuscript is kind of messy for the logical presentation. The author should adjust more to make it clearly.

Author Response

Reviewer #2:

Comment 1: In this manuscript, the author explored the functional effect of exogenous SHMP on female reproduction, especially on oocyte maturation and development. The results showed that SHMP could serve as an initiator of calcium rise and effect the fertilization process in mouse oocytes in vitro. The following are some concerns that need to be clarified.

Response: First of all, we appreciate your thoughtful comment.

Comment 2: For line 67, it is obscured and confusing for the readers to understand. What detailed effect did the author explore about exogenous SHMP supplementation? More clarifications should be addressed at the beginning (introduction part) to clearly state the scientific question of the whole manuscript.

Response: Thank you for your useful comment. Accordingly, we have inserted sentences in the introduction section (lines 66 71).

Comment 3: Line 77, may I know the reason that prepared solution need to be stored at RT for 7 days until use? Is there any purpose to deliberately do that?

Response: Thank you for your valuable comment. The SHMP solution was inactive immediately after preparation. Therefore, the SHMP solution was incubated for 2 days and longer to see whether the activity would begin to appear. We assume that the structure, shown in Figure 1a, will be being formed and stabilized 7 days after preparation. We have inserted the sentences in the materials and methods section (lines 82 – 85).

Comment 4: Since the SHMP caused partial parthenogenesis, did the author tested the location of the spindles? Was that also be affected?

Response: Thank you for your valuable comment. Accordingly, we have examined electron microscopic images again. Microfilaments, but not the spindle, showed structural changes, and its length became shorter in SHMP-treated oocytes. We have added a new Figure 4g, h and inserted sentences in the results and discussion sections (lines 275 - 277 and lines 404 - 408).

Comment 5: The paper only listed several phenotypes after the exogenous SHMP supplementation, no detailed mechanism was explored based on these findings. The author should address the limitation deeply in the discussion part for this whole manuscript.

Response: Accordingly, we have inserted new paragraphs regarding mechanisms (lines 387 – 418) and added sentences concerning limitation (lines 428 – 430).

Comment 6: I recommend the author to reconsider and adjust the title. Since there is no necessary to mention the “food additive” based on this scientific question. What’s more, more instructible information should be shown directly in the title, to help readers understand the scientific issue directly and clearly.

Response: As the reviewer pointed out, we have changed the previous title to “Sodium hexametaphosphate serves as an inducer of calcium signaling”.

Comment 7: The whole manuscript is kind of messy for the logical presentation. The author should adjust more to make it clearly.

Response: Thank for your valuable comment. We have improved the sentences throughout the manuscript.

Round 2

Reviewer 2 Report

Overall the points that I raised have been addressed properly. I agree to publish this paper at the present form.

Author Response

Thanks for your review.